# Wideband Interference Cancellation System Based on a Fast and Robust LMS Algorithm

**DOI:** 10.3390/s23187871

**Published:** 2023-09-13

**Authors:** Qiaran Lu, Huanding Qin, Fangmin He, Yunshuo Zhang, Qing Wang, Jin Meng

**Affiliations:** National Key Laboratory of Electromagnetic Energy, Naval University of Engineering, Wuhan 430033, China; lqr92138@163.com (Q.L.); q_fighting@163.com (H.Q.); hefangminemc@126.com (F.H.); gudaowork@163.com (Y.Z.); mengjinemc@163.com (J.M.)

**Keywords:** interference cancellation, variable step size, stability, steady-state error, convergence speed, interference cancellation ratio (ICR)

## Abstract

The interference cancellation ratio (ICR) is a key performance indicator of digital-to-analog hybrid radio frequency (RF) interference cancellation systems. Aiming at the low convergence speed of a digital-to-analog hybrid RF interference cancellation system based on a multi-tap structure (MDARFICS), a novel, fast, and robust variable-step-size least-mean-square (LMS) algorithm based on an improved hyperbolic tangent function (IHVSS-LMS) is proposed. The IHVSS-LMS algorithm adopts an improved hyperbolic tangent function and uses adjustable parameters and the iteration number to jointly adjust the step size, which improves the convergence speed and reduces the computational complexity. Moreover, by using the prior information of the input signal, the non-linear relationship between the step size and the input signal power is established, which enhances the robustness and the ability to suppress interference with mutable power. The IHVSS-LMS algorithm is applied to the MDARFICS. Through theoretical derivation, the convergence speed and the steady-state expressions of the interference cancellation ratio of the MDARFICS are obtained. The simulation results show that under the conditions of high and low signal-to-noise ratio (SNR), the robustness, convergence speed, and steady-state error performance of the IHVSS-LMS algorithm are better than the existing variable-step-size algorithm. The experimental results show that using the IHVSS-LMS algorithm, the MDARFICS can not only effectively accelerate the convergence speed by at least three times but can also improve the ICR by more than 3 dB.

## 1. Introduction

With the development of modern communications, the communication bandwidth has become wider, and co-time co-frequency full-duplex communications are currently a popular research area. However, interference is unavoidable. Adaptive filters are adopted for interference cancellation [1]. For wideband interference, scholars have adopted multi-tap structures to suppress it, such as digital multi-tap [2,3,4,5], analog multi-tap [6,7,8], and digital–analog hybrid multi-tap [9,10,11,12,13]. However, the taps are coupled with each other, and the matrix dimension is high, so the convergence speed of the multi-tap structure is slow.

B. Widrow studied the adaptive noise cancellation method and proposed the LMS algorithm [14]. The LMS algorithm is simple, widely used, and easy to implement. Therefore, researchers have conducted in-depth research on it [15,16]. The fixed-step-size LMS (FSSLMS) algorithm has low computational complexity and can converge to the Wiener solution. However, it cannot meet the requirements of fast convergence and low steady-state error simultaneously. Additionally, the FSSLMS algorithm would deteriorate in variable SNR scenarios [17]. Therefore, many scholars proposed the variable-step-size LMS (VSSLMS) algorithm. At the beginning of the VSSLMS algorithm, a large step size is used to speed up the convergence, and a small step size is used to reduce the steady-state error after convergence. Xiaodong Luo [18] and Babur Jalal [19] proposed the VSSLMS algorithm based on the Sigmoid, which has a fast convergence speed and low steady-state error. However, when it tended to converge, the step size would change rapidly because of the characteristics of the S-function, which increased the steady-state error. Jianwu Zhang [20] and Fuqing Tian [21] proposed an improved VSSLMS algorithm based on the hyperbolic tangent function. Based on the estimated noise power, the normalized VSSLMS algorithm [22] was proposed. The algorithms proposed in [18,19,20,21,22] adjust the step size based on the residual error, which was susceptible to noise. Jiancheng Liu [23] established the non-linear relationship between the step size and the iteration number. Its tracking ability was insufficient, and the computational complexity was large. Lu proposed a variable-step-size algorithm based on the exponential function [24], and this method uses too many exponential operations in the running process, which leads to high complexity. Based on [24], He used the variable-step-size method to update the partial filter weighting coefficients [25], which improves the convergence speed and reduces the complexity. However, in the low-SNR environment, the algorithm has a large step size when approaching convergence, so the steady-state performance is poor. Furthermore, in reference [26,27,28,29], Zhi Lin studied wireless interference suppression technology; methods such as SLNR-based secure energy-efficient beamforming were used, and the performance of satellite systems was improved effectively.

Furthermore, on co-cite platforms such as ships and aircraft, the interference power changes quickly because of power transmitting equipment such as radars, GPS, and jammers [5,30], so the LMS algorithm has to deal with a variety of fast-changing interferences [31,32]. The LMS algorithm, which adopts the hyperbolic tangent function, uses a large step size in the initial convergence stage to have a faster convergence speed and tracking speed, and after the algorithm converges, to achieve the best steady-state offset, a small step size is adopted [33,34,35]. However, the computational complexity of the hyperbolic tangent function is high, and the tracking performance is poor. The existing variable-step-size LMS algorithms are mostly used in communication scenarios with stable channels, and they are biased towards a cancellation system with a simple structure. The applicability of the variable-step-size LMS algorithms to multi-tap cancellation structures has not yet been studied.

Based on this, this paper takes the MDARFICS as the application background to study the fast variable-step-size LMS algorithm. This paper proposes the IHVSS-LMS algorithm, which adopts an improved hyperbolic tangent function, introduces adjustable parameters to control the step size, and uses the prior information of the input signal to build the non-linear relationship between the step size and the input signal. The theoretical model of the IHVSS-LMS algorithm is established, and its convergence, stability, and steady-state error performance are analyzed. Then, the IHVSS-LMS algorithm is applied to the MDARFICS, and the expressions of the convergence speed and ICR are derived. The cancellation performance of the MDARFICS is analyzed and verified. The IHVSS-LMS algorithm further improves the convergence speed, increases the ICR, and enhances noise immunity.

## 2. The Principle of the MDARFICS

The block diagram of the MDARFICS is shown in Figure 1. The analog vector modulation module simulates multipath interference by setting the delay of each tap to obtain the cancellation signal. The frequency conversion filtering module down-converts the reference signal and the residual error signal to the intermediate frequency and filters it. The digital control module converts the received signal from analog to digital, and then the digital signal is subjected to a correlation operation to obtain weights, which are used to control the amplitude attenuation and phase change of the vector modulator. The weights enter the analog vector modulation module after digital-to-analog conversion. *d_i_* is the delay of the *i*-th tap.

On platforms such as ships and aircraft, the interference of the reflection path of the high-power equipment is large, which will block or even damage the sensitive receiver on the common platform. Therefore, the influence of the multipath effect of the spatial coupling path on the cancellation performance must be analyzed. The time-domain impulse response of the spatial coupling path can be expressed as:(1)$$h\left(t\right)=\sum_{i=0}^{l}h_{i}\delta \left(t-\tau_{i}\right),$$
where *h*_0_ represents the amplitude attenuation of the direct path. *τ*_0_ is the corresponding delay, and *h_i_* (*i* = 1, 2, …,) is the amplitude attenuation of the *i*-th reflection path, and *τ_i_* is the delay of the reflection path, and *l* is the total number of reflection paths. *l* = 0 indicates that the wireless channel only consists of a direct path.

Denote the radio frequency signal radiated by the transmitter antenna of the high power equipment as *X*_S_(*t*), the interference signal entering the front end of the receiver through the spatial coupling path can be expressed as:(2)$$X_{I}\left(t\right)=X_{S}\left(t\right)*h\left(t\right)=\sum_{i=0}^{l}h_{i}X_{S}\left(t-\tau_{i}\right).$$

In the MDARFICS, denote the delay of the *i*-th tap reference signal as *d_i_*, the in-phase path and quadrature path signals obtained after the quadrature power divider are *X*_SI,*i*_(*t*) and *X*_SQ,*i*_(*t*), respectively, and the weights of the corresponding vector modulators are *W*_I,*i*_(*t*) and *W*_Q,*i*_(*t*), respectively, and then the cancellation signal can be expressed as:(3)$$Y\left(t\right)=\sum_{i=1}^{N}X_{\mathrm{SI},i}\left(t\right)W_{I,i}\left(t\right)+X_{\mathrm{SQ},i}\left(t\right)W_{Q,i}\left(t\right).$$

Denote the receiver noise as *n*(t) and the desired signal as *s*(t), and the received signal before cancellation can be expressed as:(4)$$r\left(t\right)=n\left(t\right)+s\left(t\right)+X_{I}\left(t\right).$$

The residual error signal after cancellation can be expressed as:(5)$$X_{E}\left(t\right)=r\left(t\right)-Y\left(t\right)=n\left(t\right)+s\left(t\right)+X_{I}\left(t\right)-Y\left(t\right).$$

By adjusting the weights of the in-phase and quadrature paths of each vector modulator, we can obtain the minimum residual error signal power, and the weight at this time is the optimal weight. Therefore, the optimization criterion of the cancellation parameter can be expressed as:(6)$$\begin{array}{c} {\left(W_{I,i},W_{Q,i}\right)}_{optimal}=\arg\;\underset{W_{I,i},W_{Q,i}}{\min}\;E\left\{{\left(X_{E}\left(t\right)\right)}^{2}\right\}\\ s.t.\;\left|W_{I,i}\right|\leq 1,\left|W_{Q,i}\right|\leq 1 \end{array},$$
where *E*{·} represents the expectation, and *E*{(*X*_E_(*t*))^2^} is the residual error signal power, $W_{I,i}$ and $W_{Q,i}$ are the weights of in-phase and quadrature paths of the *i*-th tap.

Since only the interference signal in the received signal is related to the reference signal, and the desired signal and receiver noise are not related to the reference signal, the residual error signal power can be obtained as:(7)$$\begin{array}{l} P_{E}=E\left\{X_{I}{}^{*}\left(t\right)X_{I}\left(t\right)\right\}+E\left\{n^{*}\left(t\right)n\left(t\right)\right\}+E\left\{s^{*}\left(t\right)s\left(t\right)\right\}\\ +K^{2}w_{i}{}^{H}Rw_{i}-Kq^{H}w_{i}-Kw_{i}{}^{H}q \end{array},$$
where *K* is the loop gain, *E*{*X*_I_^*^(*t*)*X*_I_(*t*)} is the interference signal power, *E*{*n*^*^(*t*)*n*(*t*)} is the receiver noise power, *E*{*s*^*^(*t*)*s*(*t*)} is the desired signal power, $w_{i}$ is the N×1 dimensional weight coefficient vector, **R** = *E*{***X***_S_^*^***X***_S_^T^} is the N×N dimensional reference signal autocorrelation matrix, and **q** = *E*{***X***_S_^*^*X*_I_(*t*)} is the N×1 dimensional cross-correlation vector. In general, when the number of taps is small, the matrix **R** is a non-singular matrix. It is easy to prove that **R** is a Hermitian matrix, that is, **R**^H^ = **R**. Therefore, **R**^−1^ is a Hermitian matrix, that is, (**R**^−1^)^H^ = **R**^−1^. The weights are adjusted according to the residual error signal to obtain the optimal performance.

## 3. Proposed Variable-Step-Size LMS Algorithm

The block diagram of the fixed-step LMS algorithm is shown in Figure 2. *x*(*n*) is the input signal, *ε*(*n*) is the noise, and *y*(*n*) is the output signal. The desired signal *d*(*n*) is composed of the *y*(*n*) and *ε*(*n*), and *e*(*n*) is the residual error signal, which is fed back to the adaptive filter to update the weight coefficient vector. The criterion of the adaptive filter is to minimize the residual error signal power. The mathematical model is:(8)$$y\left(n\right)=w^{H}\left(n\right)x\left(n\right),$$
(9)$$e\left(n\right)=d\left(n\right)-w^{H}\left(n\right)x\left(n\right),$$
(10)$$w\left(n+1\right)=w\left(n\right)+\mu \left(n\right)e\left(n\right)x\left(n\right),$$
where **w**(*n*) = [*w*_0_(*n*), *w*_1_(*n*), …, *w_N_*_−1_(*n*)]^T^ is weight coefficient, **x**(*n*) = [*x*(*n*), *x*(*n +* 1), …, *x*(*n + N* − 1)]^T^ is the input signal vector, *μ*(*n*) is the step size.

From [33], the sufficient convergence condition and the iteration number are:(11)$$\left\{\begin{array}{l} 0<\mu <2/\lambda_{\max}\\ n_{\chi }=\ln\left(\chi \right)/\left[2\ln\left(1-\mu \lambda_{\min}\right)\right] \end{array}\right.,$$
where *λ*_max_ and *λ*_min_ are the maximum and minimum eigenvalue of the autocorrelation matrix of the input signal, respectively. *χ* (0 < *χ* ≤ 1) is the convergence threshold, and *n_χ_* is the iteration number.

The steady-state error and offset ratio are:(12)$$\zeta =\sum_{i=0}^{M-1}\frac{\mu \lambda_{i}\sigma_{\varepsilon }{}^{2}}{2-\mu \lambda_{i}},$$
(13)$$\upsilon =\frac{\zeta }{\sigma_{\varepsilon }{}^{2}}=\sum_{i=0}^{M-1}\left(\frac{2}{2-\mu \lambda_{i}}-1\right),$$
where σ_ε_^2^ is the power of external noise. It can be seen that when the input signal and the noise signal are determined, *μ* is the key to improve the algorithm convergence speed and reduce the steady-state error.

The step size of the existing VSSLMS algorithm is controlled by the residual error signal. However, the calculation is complex, and it is easily affected by interference such as external noise.

References [20,21] proposed a variable-step-size algorithm based on the hyperbolic tangent function. To strengthen the relationship between the step size and the input signal and make the algorithm have good tracking ability, Tian added a step size feedback factor ***J***(n), whose step factor is:(14)$$\begin{array}{l} J\left(n\right)=s\cdot J\left(n-1\right)+e\left(n\right)x\left(n\right)\\ \mu \left(n\right)=\alpha (1-\frac{m+1}{m+\exp(\beta |e\left(n\right)e\left(n-1\right)|{\left\|J\left(n\right)\right\|}^{2})}) \end{array},$$
where *s* is the control coefficient.

However, the error signal and the input signal has no correlation in the initial stage of convergence in this method, so *e*(*n*)*e*(*n* − 1) will cause the algorithm to reduce the step size to the minimum value when the algorithm is not converged, which will reduce the convergence speed. In addition, the step size factor contains residual error signal, so the computational complexity is large. Jiancheng Liu [23] adopts the third-order S-function, but this structure requires two exponential calculations, which is complicated. Additionally, the convergence speed is easily affected by noise.

Therefore, this paper proposes the fast and robust variable-step-size LMS algorithm based on improved hyperbolic tangent function (IHVSS-LMS). This IHVSS-LMS algorithm only needs one exponential calculation, and the convergence speed can be controlled by adjustable parameters. Furthermore, the iterative control method is used to enhance the stability, and the non-linear relationship between the step size and the input signal is established, which enhances the ability to suppress the interference with mutable power. The mathematical model of the step size is:(15)$$\begin{array}{l} \mu \left(n\right)=\mu_{\min}+\mu_{\max}\left[1-\exp{\left(-k/\left(n^{2}+1\right)\right)}^{2\alpha +1}\right]\\ \mu \left(n+1\right)=\beta \mu \left(n\right)+\left(1-\beta \right)/\left(\gamma +\sigma_{x}{}^{2}\left(n\right)\right) \end{array},$$
where *μ*_min_ is the minimum value of step size, which can be obtained from the iteration number, and *μ*_max_ is the maximum value, which can be obtained from the steady-state error. *α*, *k,* and *β* (0 < *β* < 1) are the adjustment parameters, *σ***_x_**^2^(*n*) is the input signal power, and *γ* (0 < *γ* < 1) is a small positive number, which is added to prevent the mutation of the input signal power from causing the IHVSS-LMS algorithm to diverge.

The change curve of the step size is shown in Figure 3. It can be seen that the change curve of the step size conforms to the concave-convexity of the hyperbolic tangent function, so that the step size decreases rapidly at the beginning of the algorithm and changes slowly in the stable stage of the algorithm. As *k* increases, the convergence speed gradually increases. Parameter *k* determines the convergence speed of the step size. The larger the *k*, the faster the convergence speed.

### 3.1. Convergence of Step Size

From [36], the sufficient convergence condition is:(16)$$E\left\{\mu \left(n+1\right)\right\}<2/\lambda_{\max}.$$

Put (15) into (16), and we can obtain:(17)$$\begin{array}{l} E\left\{\mu \left(n+1\right)\right\}=\\ \;E\left\{\beta \mu \left(n\right)+\left(1-\beta \right)/\left(\gamma +\sigma_{x}{}^{2}\left(n\right)\right)\right\}<2/\lambda_{\max} \end{array}.$$

When the IHVSS-LMS algorithm tends to steady state, *μ*(*n*) changes slowly. So *μ*(*n* + 1) = *μ*(*n*), and *σ***_x_**^2^(*n*) = E{**x**(*n*)**x***^H^*(*n*)} = tr{**R**}. Then we can obtain:(18)$$E\left\{\mu \left(n\right)\right\}=1/\left(\gamma +\mathrm{tr}\left(R\right)\right),$$
where **R** = E{**x**(*n*)**x***^H^*(*n*)} is the autocorrelation matrix of the input signal, and tr() represents the trace of the matrix. Comparing (16) and (18), we can obtain that the IHVSS-LMS algorithm satisfies sufficient conditions for convergence.

### 3.2. Performance of IHVSS-LMS Algorithm

#### 3.2.1. Stability

Since the IHVSS-LMS algorithm is a negative feedback structure, the stability must be firstly ensured. We can obtain the following equation by substituting (15) into (10) and calculating the mathematical expectation.
(19)$$\begin{array}{l} E\left\{w\left(n+1\right)\right\}=\\ \;\;E\left\{w\left(n\right)\right\}\left(1-\mu \left(n\right)\sigma_{x}{}^{2}\left(n\right)\right)+E\left\{\mu \left(n\right)d\left(n\right)x\left(n\right)\right\} \end{array}.$$

Do z-transform to (19), and we can get the transfer function.
(20)$$H\left(Z\right)=z-\left(1-\mu \left(n\right)\sigma_{x}{}^{2}\left(n\right)\right).$$

If the algorithm is to be stable, all zeros and poles of the transfer function must fall within the unit circle, so:(21)$$\left|1-\mu \left(n\right)\sigma_{x}{}^{2}\left(n\right)\right|<1.$$

Substitute (15) and simplify it, and we can get:(22)$$\left|\beta \left(\mu_{\min}+\mu_{\max}\left[1-\exp{\left(\frac{-k}{{\left(n-1\right)}^{2}+1}\right)}^{2\alpha +1}\right]\right)+\frac{1-\beta }{\gamma +\sigma_{x}{}^{2}\left(n-1\right)}\right|<\frac{1}{\sigma_{x}{}^{2}\left(n\right)}.$$

When the value of n increases gradually, *μ*(*n*) changes slowly, so it can be simplified as:(23)$$\mu <\frac{2+\beta }{\beta \mathrm{tr}\left(R\right)}.$$

By combining (18) with the convergence conditions, we can get:(24)$$\lambda_{\max}/\left(2\mathrm{tr}\left(R\right)-\lambda_{\max}\right)<\beta <1.$$

The *β* affects the stability of the IHVSS-LMS algorithm and does not affect the convergence.

#### 3.2.2. Steady-State Error

Ideally, the LMS algorithm can use the input signal to completely filter out the interference signal. Therefore, the output signal only has the additive white Gaussian noise. Moreover, when the system is in a steady state, the weights do not change. Therefore, we can get the steady-state error is:(25)$$\begin{array}{l} E\left\{e\left(n\right)\right\}=\\ \;\frac{E\left\{\left(1-w^{H}\left(n\right)\right)x\left(n\right)\right\}}{1+\left(\mu_{\min}+\mu_{\max}\left[1-\exp{\left(-k/\left(n^{2}+1\right)\right)}^{2\alpha +1}\right]\right)E\left\{\sigma_{x}{}^{2}\left(n\right)\right\}} \end{array}.$$

It can be seen that when the IHVSS-LMS algorithm reaches the steady state, the steady-state error is mainly determined by *α*. The smaller α, the smaller the steady-state error.

## 4. MDARFICS Based on the IHVSS-LMS Algorithm

In our early research [13], we obtained the weight model of the multi-tap digital-analog hybrid cancellation system.

In combination with (15) we can get:(26)$$\begin{array}{l} \frac{k_{C}\beta KP_{1}}{B}\left(\mu_{\min}+\mu_{\max}\left[1-\exp{\left(\frac{-k}{n^{2}+1}\right)}^{2\alpha +1}\right]\right)\Phi {\left(d_{i}\right)}^{T}w\left(n\right)\\ +w_{i}\left(n+1\right)+k_{C}\left(1-\beta \right)\Phi_{\exp}{\left(d_{i}\right)}^{T}w\left(n\right)+F\left(n\right)=0 \end{array},$$
where *k*_C_ is the filter coefficient of the digital filter, *F*(*n*) and ***W***(*n*) have no correlation. *B* is the interference bandwidth. *P*_1_ is the transmitted signal power, and $\Phi \left(\tau \right)=E\left\{V_{S}{}^{*}\left(t\right)V_{S}\left(t+\tau \right)\right\}$, $\Phi \left(d_{i}\right)={\left[\Phi \left(d_{i}-d_{1}\right)e^{j2\pi f_{c}\left(d_{i}-d_{1}\right)},\Phi \left(d_{i}-d_{2}\right)e^{j2\pi f_{c}\left(d_{i}-d_{2}\right)},\cdots ,\Phi \left(0\right),\cdots \right]}^{T}$, $\Phi_{\exp}\left(d_{i}\right)={\left[e^{j2\pi f_{c}\left(d_{i}-d_{1}\right)},e^{j2\pi f_{c}\left(d_{i}-d_{2}\right)},\cdots ,e^{j2\pi f_{c}\left(d_{i}-d_{i}\right)},\cdots \right]}^{T}$.

The elements of matrix **R** can be obtained as:(27)$$R_{ij}=e^{j2\pi f_{c}\left(d_{i}-d_{j}\right)}\Phi \left(d_{i}-d_{j}\right).$$

The *k*-th element of the vector **q** is:(28)$$q_{k}=\sum_{i=0}^{l}\left[h_{i}e^{j2\pi f_{c}\left(d_{k}-\tau_{i}\right)}\Phi \left(d_{k}-\tau_{i}\right)\right].$$

On this basis, we will combine the IHVSS-LMS algorithm and the MDARFICS to analyze the convergence speed and ICR.

### 4.1. Convergence Speed

The weight eigenmatrix of MDARFICS is:(29)$$\left(\begin{array}{ccc} a_{11} & \ldots & a_{1N}\\ \vdots & a_{ij} & \vdots \\ a_{N1} & \cdots & a_{NN} \end{array}\right),$$
where $\left\{\begin{array}{l} a_{ij}=\left[\begin{array}{l} k_{C}\beta g\left(n\right)\Phi \left(d_{i}-d_{j}\right)\\ -k_{C}\left(1-\beta \right) \end{array}\right]e^{j2\pi f_{c}\left(d_{i}-d_{j}\right)},\;i\neq j\\ a_{ij}=k_{C}\beta g\left(n\right)\Phi \left(0\right)+k_{C}\beta -1,\;\;\;\;i=j \end{array}\right.$, $g\left(n\right)=\frac{KP_{1}}{B}\left(\mu_{\min}+\mu_{\max}\left[1-\exp{\left(\frac{-k}{n^{2}+1}\right)}^{2\alpha +1}\right]\right)$.

Because of the higher order of the weight eigenmatrix, it is difficult to solve the inverse matrix, so the theoretical expression of the optimal cancellation performance of MDARFICS is difficult to quantitatively obtain. In addition, the larger the number of taps, the more complicate the weight matrix. Therefore, it is generally only possible to quantitatively research the cancellation performance when the number of taps is small. Next, this paper takes the three-tap interference cancellation structure as an example, and quantitatively solves the theoretical expression of the convergence time required for the system to reach the stable state.

The autocorrelation matrix of the transmitted signal is as follows in (30).
(30)$$R=\left(\begin{array}{ccc} \Phi \left(0\right) & \Phi \left(d_{1}-d_{2}\right)e^{j2\pi f_{c}\left(d_{1}-d_{2}\right)} & \Phi \left(d_{1}-d_{3}\right)e^{j2\pi f_{c}\left(d_{1}-d_{3}\right)}\\ \Phi \left(d_{2}-d_{1}\right)e^{j2\pi f_{c}\left(d_{2}-d_{1}\right)} & \Phi \left(0\right) & \Phi \left(d_{2}-d_{3}\right)e^{j2\pi f_{c}\left(d_{2}-d_{3}\right)}\\ \Phi \left(d_{3}-d_{1}\right)e^{j2\pi f_{c}\left(d_{3}-d_{1}\right)} & \Phi \left(d_{3}-d_{2}\right)e^{j2\pi f_{c}\left(d_{2}-d_{1}\right)} & \Phi \left(0\right) \end{array}\right),$$

Denote $A=\left(\begin{array}{ccc} 0 & e^{j2\pi f_{c}\left(d_{1}-d_{2}\right)} & e^{j2\pi f_{c}\left(d_{1}-d_{3}\right)}\\ e^{j2\pi f_{c}\left(d_{2}-d_{1}\right)} & 0 & e^{j2\pi f_{c}\left(d_{2}-d_{3}\right)}\\ e^{j2\pi f_{c}\left(d_{3}-d_{1}\right)} & e^{j2\pi f_{c}\left(d_{3}-d_{2}\right)} & 0 \end{array}\right)$, and we can obtain the weight equation of the three-tap cancellation structure in (31).(31)$$\left(\begin{array}{c} W_{1}\left(n\right)\\ W_{2}\left(n\right)\\ W_{2}\left(n\right) \end{array}\right)=U\cdot \mathrm{diag}\left\{\frac{\left(1-e^{-\left(\begin{array}{c} k_{C}\beta g\left(n\right)\lambda_{R,i}+k_{C}\beta -1\\ +\left(k_{C}\beta -k_{C}\right)\lambda_{A,i} \end{array}\right)n}\right)}{\begin{array}{c} k_{C}\beta g\left(n\right)\lambda_{R,i}+\left(k_{C}\beta -1\right)\\ +\left(k_{C}\beta -k_{C}\right)\lambda_{A,i} \end{array}}\right\}U^{H}\left[k_{C}\beta h_{0}g\left(n\right)\left(\begin{array}{c} \Phi \left(d_{1}-\tau_{0}\right)e^{j2\pi f_{c}\left(d_{1}-\tau_{0}\right)}\\ \Phi \left(d_{2}-\tau_{0}\right)e^{j2\pi f_{c}\left(d_{2}-\tau_{0}\right)}\\ \Phi \left(d_{3}-\tau_{0}\right)e^{j2\pi f_{c}\left(d_{3}-\tau_{0}\right)} \end{array}\right)+\left(\begin{array}{c} k_{C}\left(1-\beta \right)h_{0}e^{j2\pi f_{c}\left(d_{1}-\tau_{0}\right)}\\ k_{C}\left(1-\beta \right)h_{0}e^{j2\pi f_{c}\left(d_{2}-\tau_{0}\right)}\\ k_{C}\left(1-\beta \right)h_{0}e^{j2\pi f_{c}\left(d_{3}-\tau_{0}\right)} \end{array}\right)\right]$$

The matrix **R** can be decomposed diagonally into **R** = **UΛ**_R_**U**^H^. **Λ**_R_ = diag{*λ*_R,*i*_}, *i* = 1, 2, 3 is the eigenvalue of the autocorrelation matrix of the reference signal, and the eigenvalue of the matrix A is **Λ**_A_ = diag{*λ*_A,*i*_}.

Then the convergence time of *W*(n) is:(32)$$t_{\mathrm{conv}}=\frac{B}{\left(\begin{array}{l} k_{C}\beta KP_{1}\left(\mu_{\min}+\mu_{\max}\left[1-\exp{\left(\frac{-k}{n^{2}+1}\right)}^{2\alpha +1}\right]\right)\lambda_{R,i}\\ +\left(k_{C}\beta -k_{C}\right)\lambda_{A,i}+\left(k_{C}\beta -1\right) \end{array}\right)},$$

We can see that the parameters *k*_C_, *α*, *k,* and *β* jointly affect the convergence time. *k*_C_ and *β* mainly control the stability, parameter *k* can effectively improve the convergence speed, and parameter *α* can optimize the convergence speed and steady-state error performance. Therefore, the proposed IHVSS-LMS algorithm can improve the convergence speed by reasonably adjusting the values of *k* and *α*.

### 4.2. ICR

The ICR is usually used to characterize the cancellation capability of interference cancellation, which is defined as the ratio of the received signal before cancellation to the residual error signal power after cancellation.

In early research [13], the power of the interference signal is:(33)$$P_{I}=\frac{P_{1}}{B}\sum_{i=1}^{l}\sum_{j=1}^{l}Bh_{i}h_{j}\sin c\left[B\left(\tau_{i}-\tau_{j}\right)\right]e^{-j2\pi f_{c}\left(\tau_{i}-\tau_{j}\right)}.$$

The power of the residual error signal is:(34)$$P_{E}=\frac{P_{1}}{B}\left(\begin{array}{l} \sum_{i=1}^{l}\sum_{j=1}^{l}Bh_{i}h_{j}\sin c\left[B\left(\tau_{i}-\tau_{j}\right)\right]e^{-j2\pi f_{c}\left(\tau_{i}-\tau_{j}\right)}\\ -\mu_{D}Kq^{H}R^{-1}q+{\left(\mu_{D}Kw-R^{-1}q\right)}^{H}R\left(\mu_{D}Kw-R^{-1}q\right) \end{array}\right).$$

Then the ICR of the MDARFICS can be expressed as:(35)$$ICR=\frac{\sum_{i=1}^{l}\sum_{j=1}^{l}Bh_{i}h_{j}\sin c\left[B\left(\tau_{i}-\tau_{j}\right)\right]e^{-j2\pi f_{c}\left(\tau_{i}-\tau_{j}\right)}}{\left(\begin{array}{l} \sum_{i=1}^{l}\sum_{j=1}^{l}Bh_{i}h_{j}\sin c\left[B\left(\tau_{i}-\tau_{j}\right)\right]e^{-j2\pi f_{c}\left(\tau_{i}-\tau_{j}\right)}\\ -Gq^{H}R^{-1}q+{\left(Gw-R^{-1}q\right)}^{H}R\left(Gw-R^{-1}q\right) \end{array}\right)},$$
where $G=\left(\mu_{\min}+\mu_{\max}\left[1-\exp{\left(\frac{-k}{n^{2}+1}\right)}^{2\alpha +1}\right]\right)KP_{1}$.

As the step size increases, the ICR of the system gradually increases. When the step size changes slowly, the system reaches a steady state, and the system cancellation reaches the maximum value. Considering the gain of the MDARFICS, when the weight takes the optimal value, the ICR is the largest. The IHVSS-LMS algorithm can make the MDARFICS obtain a higher ICR in the initial stage of cancellation by setting the maximum and minimum values of the step size, and in the steady stage, the ICR remains basically unchanged.

### 4.3. Complexity Analysis

Computational complexity is an important factor that affects the application of algorithms. In this section, the computational complexity of the algorithm proposed in this paper is analyzed and compared with the FXSSLMS and VSSLMS algorithms.

The algorithm in this paper requires two additions and two multiplications to calculate the step factor each time. The recursive method of the algorithm in this paper is the same as that of the FXSSLMS algorithm except for calculating the step factor—if the filter order in the algorithm is assumed to be N, the recursive operations of the algorithm in this paper require a total of 2N + 1 additions and 2N + 4 multiplications. Similarly, the computational complexities of the FXSSLMS algorithm and the VSSLMS algorithm in references [7,9,10,11] are shown in Table 1. It can be seen that the computational complexity of the algorithm in this article is slightly higher than the FXSSLMS algorithm, but lower than other VSSLMS algorithms. The algorithm in this paper has low computational complexity and is easy to implement in hardware architecture.

## 5. Simulations and Discussions

### 5.1. Analysis of the Influence of Parameters on the IHVSS-LMS Algorithm

This section verifies the analysis of *k*, *β*, and *α*; simulates the impact on the performance of the IHVSS-LMS algorithm; and selects the optimal parameters. The order of the adaptive filter is *N* = 5. The number of sampling points is 1000, assuming that the coefficient [23] of the unknown system is [0.227, 0.460, 0.688, 0.460, 0.227]^T^, and the coefficient is set to [−0.298, 0.225, 0.849, 0.225, −0.298]^T^ when the iteration number is 500. **x**(*n*) and *ε*(*n*) are zero-mean Gaussian white noise with zero mean value, and their variance is 1 and 0.001, respectively. The SNR is 30 dB. We set *μ*_max_ = 0.8(*σ***_x_**^2^/*N*) = 0.16 to ensure convergence and *μ*_min_ = 0.005(*σ***_x_**^2^/*N*) = 0.001 to reduce the steady-state error. Each learning curve is the statistical average result after 200 independent simulations. The learning curves concerning the parameters *α*, *k,* and *β* are shown in Figure 4.

Figure 4a shows the learning curves when *α* = 1, *β* = 0.98, and *k* is 10, 100, 500, 1000, and 2000, respectively. We can find that as *k* increases, the curve becomes steeper and the convergence speed becomes faster, that is, *k* controls the convergence speed. Therefore, in order to decrease the calculation amount, we take *k* as 1000.

Figure 4b shows the learning curves when *α* = 1, *k* = 1000, and *β* is 0.8, 0.9, and 0.98, respectively. We can observe that the convergence speed of the IHVSS-LMS algorithm remains unchanged during the process of increasing *β* from 0.8 to 0.98. The *β* only affects stability and does not affect the convergence speed. This is consistent with the theoretical analysis. In the practical engineering applications, we need to make *β* meet (24).

Figure 4c shows the learning curves when *k* = 1000, *β* = 0.98, and *α* is 1, 2, and 3, respectively. In the process of reducing *α* from 3 to 1, not only is the convergence speed improved but the steady-state error is reduced by about 5 dB. This shows that *α* affects the convergence speed and steady-state error, which is consistent with (25). When *α* is 1, the convergence speed and the steady-state error performance of the IHVSS-LMS algorithm are optimal.

### 5.2. Robustness Analysis of the IHVSS-LMS Algorithm

By setting different weight coefficients in different iteration numbers, the tracking ability and robustness are simulated. We use the mutation of weights to simulate that of interference power, as shown in Table 2. The sampling point is 2000, and the parameters of step size are set to *α* =1, *k* = 1000, and *β* = 0.98. The SNR is 10 dB, 20 dB, and 30 dB, respectively. The learning curves under different SNRs are shown in Figure 5.

When the IHVSS-LMS algorithm reaches the steady state, even though the parameters change, the IHVSS-LMS algorithm can still converge to the steady state quickly, so the IHVSS-LMS algorithm has good tracking ability. Under the condition of low SNR, the algorithm can still maintain good performance, so the IHVSS-LMS algorithm has good robustness.

### 5.3. Comparison of Different Algorithms of the MDARFICS

We adopt the interference cancellation system of a three-tap structure to compare the performance of different algorithms. The MDARFICS includes three vector modulators, three analog-to-digital converters (ADCs), three digital-to-analog converters (DACs), and one field-programmable gate array (FPGA). Under the same simulation conditions, taking the SNR of 30 dB as an example, the MDARFICS parameters in Table 3 and the algorithms parameters in Table 1 are used to verify the convergence speed of different algorithms, and the results are shown in Figure 6.

Comparing the four algorithms in the references, the convergence speed of the IHVSS-LMS algorithm proposed in this paper and the algorithm proposed in [23] are significantly faster than the other three variable-step-size algorithms. Compared with the variable-step-size algorithm proposed by Liu Jiancheng, the IHVSS-LMS algorithm is slightly faster. It completes the cancellation in about 50 μs, and the residual signal power is lower. The steady-state residual error performance is optimized by 3~5 dB. Therefore, the IHVSS-LMS algorithm can not only improve the convergence speed but can also improve the ICR by 3~5 dB.

## 6. Experiment Verification

In this section, the MDARFICS is used as an example to verify the superiority of the proposed IHVSS-LMS algorithm. In addition, in order to facilitate the characteristic analysis of the MDARFICS, the experimental testbed is built in a laboratory environment. The testbed is shown in Figure 7. In the same experimental scenario, the optimal parameters of different algorithms are selected for verification. The parameters of the cancellation system are shown in Table 3, and the algorithm’s parameters are in Table 1. From Table 1, we can see that the complexity of the IHVSS-LMS algorithm is lower than other algorithms.

### 6.1. Influence of Algorithm Parameters on the MDARFICS

Firstly, the influence on the convergence speed is simulated by changing different parameters in the IHVSS-LMS algorithm. The characteristics of the convergence time when the parameters *k* and *α* are changed are shown in Figure 8. Among them, *β* = 0.98 and *α* = 1 (Figure 8a), and *β* = 0.98 and *k* = 1000 (Figure 8b).

It can be seen from Figure 8a that when parameter *k* increases from 10 to 2000, the convergence time decreases from 400 μs to 200 μs. When *k* is 1000 and 2000, the convergence speed is almost the same, which indicates that *k* mainly controls the convergence speed. From Figure 8b, we can see that when parameter *α* is reduced from 3 to 1, the convergence speed is almost unchanged, but the steady-state residual error is reduced by about 3~5 dB. This indicates that *α* affects the steady-state residual error. Therefore, the experimental results are consistent with the theoretical analysis.

### 6.2. Convergence Speed

Then, we take the SNR as 30 dB and use the parameters in Table 1 to verify the convergence speed of the cancellation system under different algorithms. The result is shown in Figure 9.

It can be seen that the convergence speed of the system with adopting the IHVSS-LMS algorithm is faster than other algorithms. All the convergence time of the algorithms in [19,21,22] exceed 600 μs, and the convergence time of the algorithm in [23] is about 180 μs. However, the convergence speed of the MDARFICS with IHVSS-LMS algorithm is faster than the other four algorithms, and the convergence time is about 50 μs. Therefore, the convergence speed is accelerated more than three times. Moreover, the steady-state performance of IHVSS-LMS algorithm is apparently better than the other algorithms. Therefore, the proposed IHVSS-LMS algorithm can not only improve the convergence speed but it can also improve the ICR of the system.

### 6.3. ICR

It can be seen from the above results that the algorithm in [23] is close to the proposed IHVSS-LMS algorithm in terms of convergence speed and steady-state error performance. Therefore, in this section, we compare the ICR performance between the IHVSS-LMS algorithm and the algorithm in [23]. The result is shown in Figure 10.

We can see that the interference signal power before cancellation is −10 dBm, and the ICR of the MDARFICS is 33 dB when the algorithm in [23] is adopted. When using the IHVSS-LMS algorithm, the ICR is 38 dB. Therefore, the steady-state error performance of the MDARFICS adopting the IHVSS-LMS algorithm is 5 dB better than the algorithm in [23], which proves the superiority of the IHVSS-LMS algorithm.

Next, by changing the transmitter power, the interference cancellation performance under different SNRs is verified. The result is shown in Figure 11.

We can observe that when the transmitter power increases from −15 dBm to 20 dBm, the SNR gradually decreases, and the residual error signal obtained by the two algorithms does not change greatly. So, the two algorithms have a large dynamic range of interference cancellation and better anti-noise performance. However, the IHVSS-LMS algorithm is 3~5 dB better than the algorithm in [23].

## 7. Conclusions

In this paper, the novel IHVSS-LMS algorithm is proposed. The IHVSS-LMS algorithm adopts an improved hyperbolic tangent function and introduces adjustable parameters to adjust the step size with the iteration number. The IHVSS-LMS algorithm is applied to the MDARFICS, and the expressions of the convergence speed and ICR are derived. The cancellation performance of the MDARFICS is analyzed and verified. The main conclusions are the following.

Compared with the four existing VSSLMS algorithms, the IHVSS-LMS algorithm has a faster convergence speed, lower steady-state error, and stronger robustness. The IHVSS-LMS algorithm has low computational complexity.The IHVSS-LMS algorithm is applied to the MDARFICS. The convergence speed can be improved by adjusting parameter *k* of the IHVSS-LMS algorithm, and the steady-state performance can be improved by adjusting parameter *α*. Compared with existing algorithms, the IHVSS-LMS algorithm can improve the convergence speed of the MDARFICS by at least three times and the ICR by more than 3 dB.

The IHVSS-LMS algorithm is of much practical application value to hardware implementations.

## Figures and Tables

**Figure 1 sensors-23-07871-f001:**
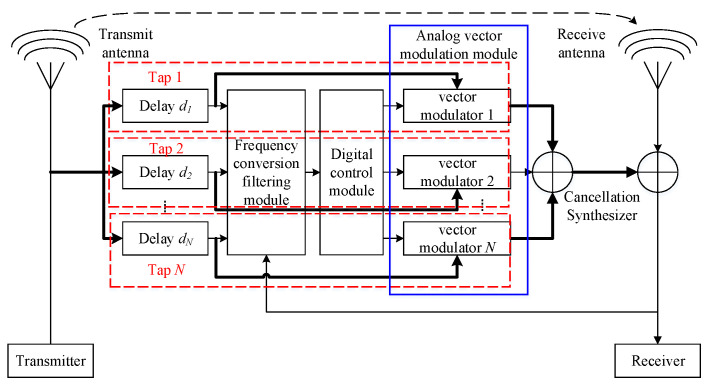
Block diagram of the MDARFICS.

**Figure 2 sensors-23-07871-f002:**
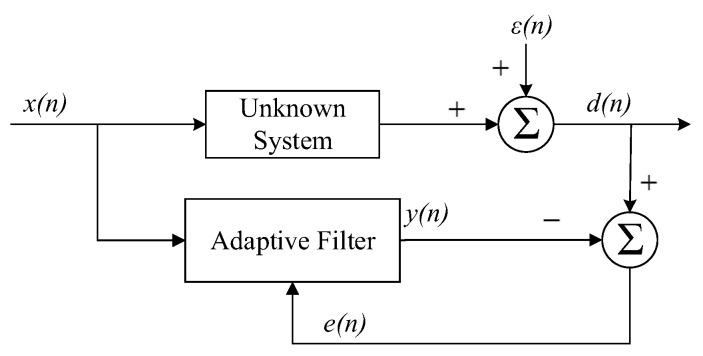
The block diagram of the fixed-step LMS algorithm.

**Figure 3 sensors-23-07871-f003:**
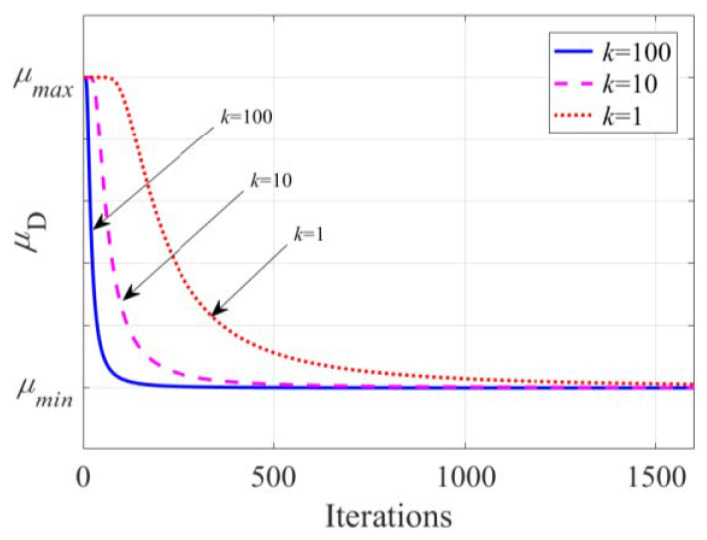
Curve of the step size.

**Figure 4 sensors-23-07871-f004:**
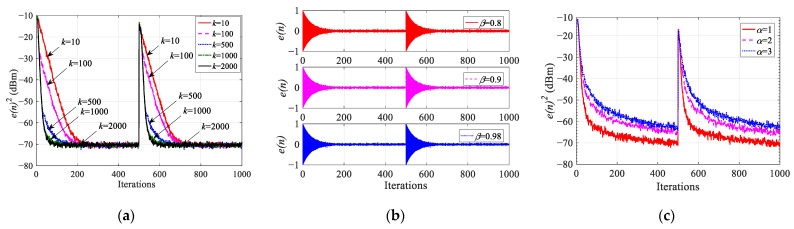
Learning curves concerning the parameters *k*, *β*, and *α*. (**a**) Effect of *k.* (**b**) Effect of *β.* (**c**) Effect of *α*.

**Figure 5 sensors-23-07871-f005:**
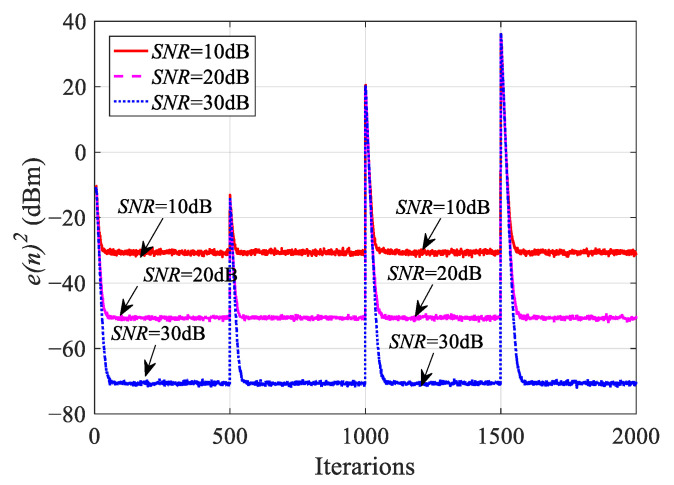
The learning curves under different SNRs.

**Figure 6 sensors-23-07871-f006:**
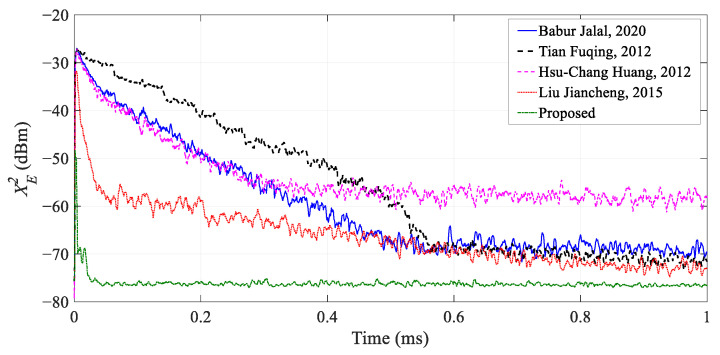
Simulation results for convergence speed under different algorithms [19,21,22,23].

**Figure 7 sensors-23-07871-f007:**
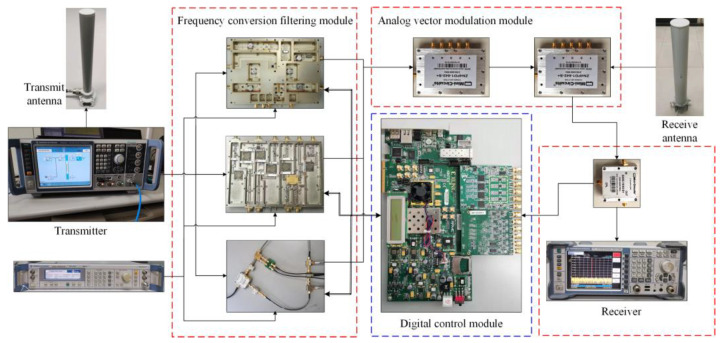
Testbed of the multi-tap digital–analog hybrid interference cancellation system.

**Figure 8 sensors-23-07871-f008:**
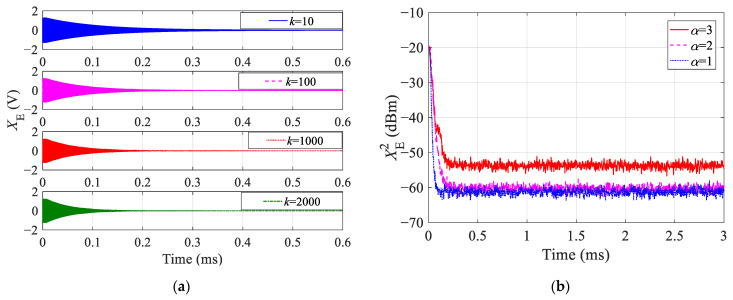
The curves of the convergence time. (**a**) Convergence curve for parameter k. (**b**) Convergence curve for parameter α.

**Figure 9 sensors-23-07871-f009:**
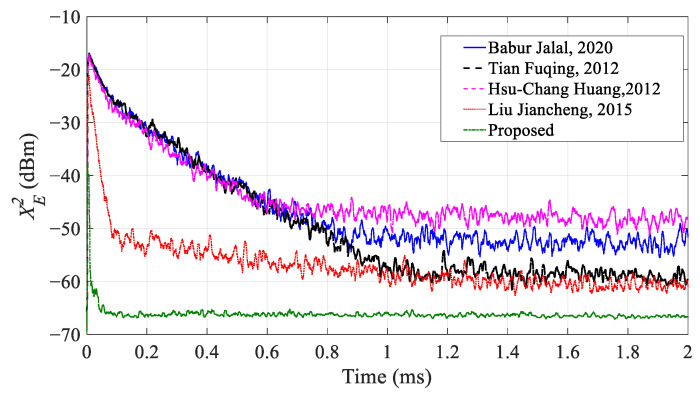
Experimental results for convergence speed under different algorithms [19,21,22,23].

**Figure 10 sensors-23-07871-f010:**
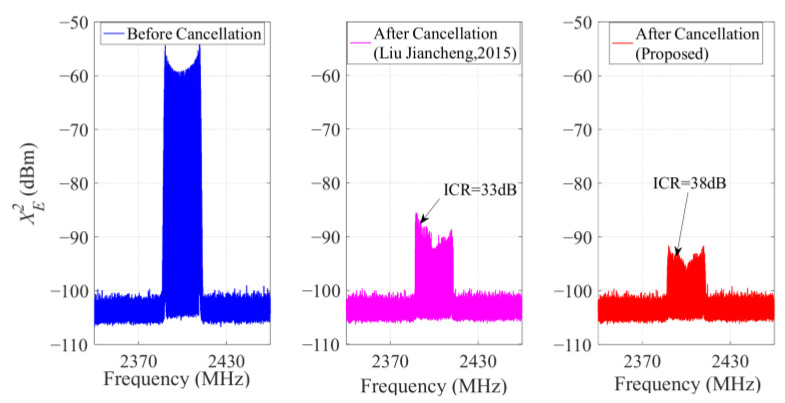
Cancellation results of the two algorithms [23].

**Figure 11 sensors-23-07871-f011:**
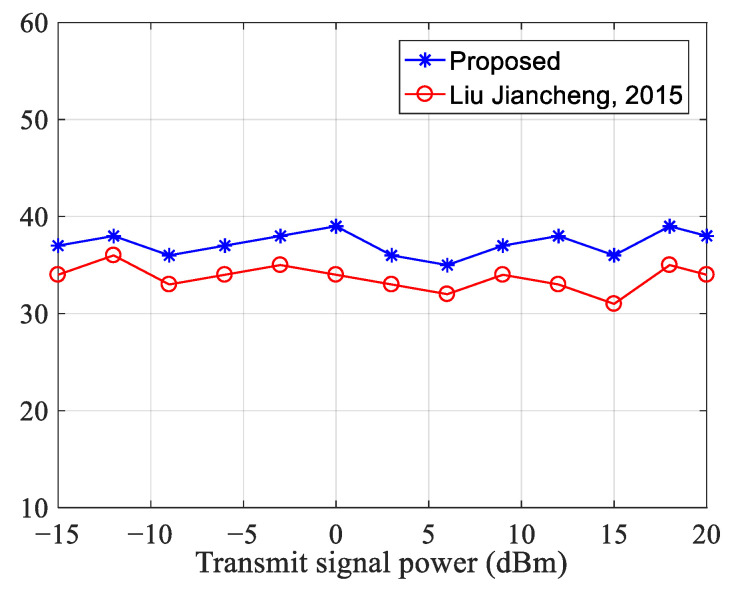
Interference suppression performance comparison [23].

**Table 1 sensors-23-07871-t001:** Simulation Parameters of Algorithms.

Algorithms	Step Size	Parameters	SNR (30 dB)	Multiplication	Additions
[19]	$\delta_{e}\left(n\right)=\delta_{e}\left(n-1\right)+\left|e\left(n\right)\right|,\sigma_{x}{}^{2}\left(n\right)=x^{H}\left(n\right)x\left(n\right),\mu \left(n\right)=\left(e^{{\tilde{\delta }}_{e}\left(n\right)}/\left(e^{{\tilde{\delta }}_{e}\left(n\right)}+1\right)\right){\left(\delta_{e}{}^{2}\left(n\right)+\sigma_{x}{}^{2}\left(n\right)\right)}^{-1}$	θ	0°	2*N* + 5	2*N* + 4
[21]	$p\left(n\right)=\exp\left[\beta \left(\left|e\left(n-1\right)e\left(n\right)\right|+0.01\left|e\left(n\right)\right|\right)\right],\mu \left(n\right)=\alpha \left(1-\left(m+k\right)/\left(m+kp\left(n\right)\right)\right)$	$\begin{array}{l} \alpha ,\beta \\ k,m \end{array}$	0.2, 1000 100, 1000	2*N* + 3	2*N* + 8
[22]	$\begin{array}{l} r_{ex}\left(n\right)=\alpha r_{ex}\left(n-1\right)+\left(1-\alpha \right)x\left(n\right)e\left(n\right),\sigma_{x}{}^{2}\left(n\right)=\alpha \sigma_{x}{}^{2}\left(n-1\right)+\left(1-\alpha \right)x^{2}\left(n\right)\\ \sigma_{\varepsilon }{}^{2}\left(n\right)=\sigma_{e}{}^{2}\left(n\right)-1/\sigma_{x}{}^{2}\left(n\right)+r_{ex}{}^{T}\left(n\right)r_{ex}\left(n\right),\mu \left(n\right)=\alpha \mu \left(n-1\right)+\left(1-\alpha \right)\sigma_{e}{}^{2}\left(n\right)/\beta \sigma_{\varepsilon }{}^{2}\left(n\right) \end{array}$	α,β	0.99, 30	3*N* + 4	5*N* + 12
[23]	$\mu \left(n\right)=\mu_{\min}+\mu_{\max}\frac{e^{{\left(k/\left(n-m+1\right)\right)}^{3}}-1}{e^{{\left(k/\left(n-m+1\right)\right)}^{3}}+1}$	k	85	2*N* + 4	2*N* + 6
Proposed	$\mu \left(n+1\right)=\beta \left(\mu_{\min}+\mu_{\max}\left[1-\exp{\left(\frac{-k}{n^{2}+1}\right)}^{2\alpha +1}\right]\right)+\frac{1-\beta }{\gamma +\sigma_{x}{}^{2}\left(n\right)}$	α,k,β	0.8, 100, 0.98	2*N* + 4	2*N* + 1

**Table 2 sensors-23-07871-t002:** Weight Coefficients.

Iteration Number	Weight Coefficient
0	[0.227, 0.460, 0.688, 0.460, 0.227]^T^
500	[−0.298, 0.225, 0.849, 0.225, −0.298]^T^
1000	[1.49, −1.125, −4.245, −1.125, 1.49]^T^
1500	[−2.98, 2.25, 8.49, 2.25, −2.98]^T^

**Table 3 sensors-23-07871-t003:** MDARFICS Parameters.

Parameter	Value
Tap number	3
Transmitter power	−10 dBm
Frequency	2.4 GHz
Interference signal bandwidth	30 MHz
ADC bit	14 bit
DAC bit	16 bit

## Data Availability

The data presented in this study are available on request from the corresponding author.

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
