# Peer review of "Wideband Interference Cancellation System Based on a Fast and Robust LMS Algorithm"

_sensors, 2023, doi:10.3390/s23187871_

Round 1

Reviewer 1 Report

Please see the attached comments.

Minor editing of English language is needed.

Reviewer 2 Report

The presented paper introduces a multi-tap structure (MDARFICS) to address the slow convergence speed of a digital-to-analog hybrid RF interference cancellation system. The results show an improved convergence speed and reduced computational complexity. These improvements reinforce the practical viability of the proposed approach. The paper is well-written and presented. Minor suggestions for this paper are

1.      The improvements achieved in convergence speed and interference cancellation might come with trade-offs in other aspects of system performance, such as stability, steady-state behavior, or adaptability to rapid changes in the input signal. This needs to be discussed.

2.      Improve figure 7, as the current one is a bit blur. 
